# Modified Rice Straw Enhanced Cadmium (II) Immobilization in Soil and Promoted the Degradation of Phenanthrene in Co-Contaminated Soil

**DOI:** 10.3390/ijms20092189

**Published:** 2019-05-03

**Authors:** Ali Mohamed Elyamine, Mohamed G Moussa, Javaria Afzal, Muhammad Shoaib Rana, Muhammad Imran, Xiaohu Zhao, Cheng Xiao Hu

**Affiliations:** 1Key Laboratory of Arable Land Conservation (Middle and Lower Reaches of Yangtze River), Ministry of Agriculture, Research Center of Micro-elements, College of Resource and Environment, Huazhong Agricultural University, Wuhan 430070, China; elyoh@hotmail.fr (A.M.E.); MohamedGomaa_Ali@agr.asu.edu.eg (M.G.M.); juvaria_afzal@outlook.com (J.A.); muhammadshoaib@webmail.hzau.edu.cn (M.S.R.); imrangorayauaf@yahoo.com (M.I.); xhzhao@mail.hzau.edu.cn (X.Z.); 2Hubei Provincial Engineering Laboratory for New Fertilizers, Huazhong Agricultural University, Wuhan 430070, China; 3Faculty of Science and Technology, Department of Life Science, University of Comoros, Moroni 269, Comoros; 4Soil and Water Research Department, Nuclear Research Center, Egyptian Atomic Energy Authority, Abou Zaabl 13759, Egypt

**Keywords:** alkali modified straw, bio-absorbent, phenanthrene metabolism, bio-immobilization, bioavailability, enzyme-chemical interaction

## Abstract

Very limited information is available about heavy metal-polycyclic aromatic hydrocarbons (PAHs) depollution involving the modified natural material in soil. Using phenanthrene and cadmium (Cd) as model, this study investigated the effect(s) of modified rice straw by a NaOH solution and on PAHs, heavy metal availability, and their interactions. Treatment included chemical contaminant with/without modified/unmodified rice straw. Fourier Transform Infrared (FTIR) analysis revealed that certain functional groups including anionic matters groups, which can a complex with Cd^2+^, were exposed on the modified rice straw surfaces. Therefore, Cd concentration was significantly reduced by about 60%, 57%, 62.5 %, and, 64% in the root, shoot, CaCl_2_, diethylenetriaminepentaacetic acid (DTPA), and extractable Cd, respectively. Subsequently, the prediction of the functional profile of the soil metagenome using Clusters Orthologous Groups (COGs) and the Kyoto Encyclopedia of Genes and Genomes (KEGG) database revealed that the significantly changed individual COGs belonged to the carbohydrate metabolism, ion transports, and signaling (including cytochrome P450s) categories. This indicated that ion transports might be involved in Cd management, while carbohydrate metabolism, including bisphenol, benzoate, ethylbenzene degradation, and cytochrome P450s, were rather involved in phenanthrene metabolism. The exposed functional group might serve as an external substrate, and P450s might serve as a catalyst to activate and initiate phenanthrene metabolism process. These finding offer confirmation that modified straw could promote the reduction of heavy metal and the degradation of PAHs in soil.

## 1. Introduction 

Since the industrial revolution, the contamination of agricultural soils with heavy metals and polycyclic aromatic hydrocarbons (PAHs) has increasingly become a serious global environmental concern and poses a huge threat to human beings and natural ecosystems. Cadmium (Cd) is one of the most known toxic pollutants in toxicological research [1]. Phenanthrene is the simplest aromatic hydrocarbon with three-fused ring compound [2] and contains both “bay-region and K-region epoxides” which are highly reactive both chemically and biologically [3]. Both chemicals negatively affect not just plant physiology and biology [4,5,6] and soil microorganisms [7,8], they also pose health risks for both humans and animals [6] through food-chain bioaccumulation [9].

Diverse approaches such as chemical precipitation, thermal processes, and physical separation, have been developed and made available to clean up the contaminated environment. However, these techniques not only present several disadvantages, such as the destruction of particle structure and microbial activities in soils, as well as high cost [10,11,12], they also, in many cases, transfer the pollutant from one form to another. Phytoremediation, which uses metal hyperaccumulator plants to remove contaminants from soils, is another cost-efficient and environmentally friendly remediation technology; however it has faced some challenges as well [13,14], such us the requirement of full understanding about how the pollutant is managed within plants at both physiological and genetic levels [15]. 

Microbial degradation is the major technique mostly used to remove, alter, or isolate PAH [16]. Microbes are well-known for their ability to catabolize and breakdown the organic compound into less complex metabolites [16] or into inorganic minerals such as H_2_O and CO_2_ (under aerobic conditions) [17] or CH_4_ (under anaerobic conditions) [3]. Therefore, a significant number of phenanthrene-degrading bacteria and fungi have been identified and isolated. However, based on the mechanism by which most of larger molecular weight PAHs such as phenanthrene are degraded, this process requires the addition of external substrates such as bicarbonate molecules to activate their metabolism [17,18] or the hydrogenation of the aromatic ring (anaerobic metabolism) [3]. In addition, considering the fact that the inoculants of exogenous/endogenous microorganisms must be able to overwhelm biotic and abiotic stress in the environment, the use of a free microorganism for bioremediation of contaminated sites in a large scale can fail. 

Many studies have thoroughly invested the interactions between PAHs and heavy metals, their individual effects on the soil organisms and plants toxicity, and different methods for the control and management their respective effects [19,20]. Though Cd is everlasting, its availability (Cd^2+^) may decrease in the environment by (1) a chelating process via chemical or physical remediation or (2) by shifting their valence by redox reaction by interacting in some specific cases with PAHs [20]. However, to the best of our knowledge, this alternative is less exploited for the purpose of PAHs-heavy metal environmental depollution, and no information is available about their depollution involving the modified natural material, such as straw, in soil.

Thus, natural materials that are available in large quantity, such as rice straw, may have potential as inexpensive sorbents or immobilizers for both organic and inorganic pollutants. As a heterogeneous material, rice straw has been used as a biosorber to reduce heavy metals in aquatic environments [21] and as a chelator source of inorganic pollutants in soil [22,23]. Agricultural crop residues, besides from the basic constituents, usually contain extractives and diverse molecules composed of a variety of functional groups capable of binding heavy metal [24]. The metals cations have a good affinity with the anionic matters present on the rice straw surfaces and can form stable complexes [25]. Subsequently, the use of organic fertilizers was shown to have positive effects on the removal of PAH in the soil [26]. Indeed, there is a close correlation between soil microbial community and the degradation of pollutants [27]. In addition, the micro-fauna densities attached to plant residues have increased with the decomposition of organic matter through the mineralization and the humification effect of microorganisms [28,29].

In the present study, phenanthrene and Cd were used as models of organic and inorganic pollutants to investigate the effect(s) of modified rice straw on PAHs and heavy metal availability, as well as their interaction. Rice straw was treated with a NaOH solution to produce carbonaceous molecules for the PAHs’ degradation activation and to provide more binding sites for Cd^2+.^ This paper aims to examine the effect of this modified material on (1) Cd and phenanthrene availability; (2) microbial functional profile prediction; and (3) phenanthrene degradation.

## 2. Results

### 2.1. Characterization of Rice Straw by FTIR 

To understand the change sequence of the functional groups of treated and not-treated rice straw, FTIR analysis was performed in the fingerprint region of the absorbance spectra (Figure 1). The characteristics absorption bands and different functional group corresponded to different peaks in Figure 1 are listed in the Table 1. According to previous literature [22,24], the bands at approximately 1640–1500 cm^−1^ correspond to either the acetyl or uranic ester groups of the hemicelluloses or the ester linkage of carboxylic group of the ferulic and p-coumaric acids of lignin and/or hemicelluloses. The bands at 1450 and 1407 cm^−1^ represent the aromatic skeletal stretching vibration C=C of lignin. On the other hand, the bands at 1242–1162 cm^−1^ can be ascribed to the C-O-C stretching vibration in aryl-alkyl ether, the C-O-C asymmetric stretch vibration, or C-O stretching and C-O deformation in cellulose and hemicellulose. The band at approximately 1321 cm^−1^ has been attributed to the CO− symmetric stretching of deprotonated ester, and the band at and 1302 cm^−1^ has been ascribed to the OH deformation of phenolic OH. All cross-peaks were positive, implying that signals of functional groups activated by NaOH had simultaneous intensity changes with each other in the same direction. The bands at 2919 and 2853 cm^−1^ were ascribed tp the C−H stretching vibration of aliphatic compounds. Though the characteristic peaks of both spectra SNT (straw not treated) and ST (treated straw) present a very similar profile from 1268–3440 cm^−1^, there were frequency shifts (intensity changes) between both rice straw samples. As shown in the Figure 1, the highest change in intensity was observed after 1268 cm^−1^. The absorptions at 1268–1039 cm^−1^ attributed to the silica (Si-O bending and Si-O stretching) decreased considerably. This indicated that a portion of silica was removed after the treatment of rice straw. The changed profile of the absorption at 1039–400 cm^−1^ for ST could suggest that after chemical treatment and the removal of silica, the groups C-O, C-O-H, C-H, and O-H became more exposed in the material.

### 2.2. Cd and Phenanthrene Determination 

#### Cd Determination in Plant

The mean concentrations of Cd in the plant root and shoot in different treatments were investigated and are plotted in Figure 2. In the treatment with Cd only, although the application of rice straw significantly decreased Cd concentration in the root (Figure 2A) and shoot (Figure 2B) compared to Cd alone, the reduction was particularly marked in the modified rice straw inoculated treatment (*p* < 0.01). The comparison of Cd concentration in plant tissues with the available Cd concentration (Cd^2+^) in different soil treatments was found to have a positive correlation. However, Cd accumulation (plant dry weight × Cd concentration) was found to decrease in the root by about 43% and 55%, respectively, in Cd applied to non-treated straw (CdSNT) and to treated straw (CdST), compared to Cd alone. Further, in the presence of phenanthrene (PCd) treatment, Cd concentration uptake by the plant was significantly reduced by about 41%, *p* = 0.027, compared to the treatment with Cd alone. Cd accumulation recorded in the root was reduced by about 53%, *p* = 0.0024, and 65%, *p* = 0.0018, in the unmodified (PCdNST) and modified (PCdST), respectively. Cd concentration in the shoot was significantly decreased by 38% and 52% in CdSNT and CdST, respectively, compared to Cd alone treatment (Figure 2B). However, in PCd treatment, the reduction was by about 43% compared to Cd treatment alone. Furthermore, 58% and 62% of Cd concentration was reduced, respectively, in PCdSNT and PCdST, compared to Cd alone. Considering the total Cd accumulated in the root and shoot, the sequence of Cd accumulation follows this trend: Cd alone > PCd > CdSNT > Cd ST > PCdNST > PCdST, with 1.708, 1, 0.98, 0.78, 0.77, and 0.588 mg Cd kg^−1^ DW, respectively. This result highlights the fact that the modified rice straw inoculation significantly decreases the total Cd reaching the plant; however, the amount of Cd accumulated in the presence of phenanthrene was much more reduced as compared to that of Cd alone

### 2.3. Cd Extraction in Soil

#### 2.3.1. CaCl_2_ Extractable Cd

Rice straw application significantly (*p* < 0.05) reduced Cd concentration in the CaCl_2_-extractation soil fraction compared to the control (Figure 3A). In the treatment with Cd only, the concentrations of Cd in pore water were significantly lower when rice straw was applied as a modified material contrary to CdSNT. Furthermore, in the presence of phenanthrene, the Cd concentration of the CaCl_2_-extractable was 0.6-fold reduced, compared to Cd alone. Though rice straw application was more effective in reducing the pore water Cd^2+^ concentration, the modified material was found to be the most effective. The treatment with unmodified material reduced pore water Cd^2+^ concentration by 41% and 59%, respectively, in CdSNT and PCdSNT, while the reduction observed in the modified material (CdST) and (PCdST) treatment was 55% and 66%, respectively. 

#### 2.3.2. Diethylenetriaminepentaaacetic Acid (DTPA)-Extractable Cd

Plant-available Cd significantly decreased with the addition of rice straw, and the effect was more pronounced in the modified materials—either in single Cd treatment (CdST) or in the mixture treatment (PCdST) (Figure 3B). The concentration of plant-available Cd in single Cd treatment was more reduced (*p* < 0.01) in the modified rice straw application as compared to the unmodified one. In the mixture treatment, despite the observed reduction in the diethylenetriaminepentaacetic acid (DTPA)-extractable Cd, no significant difference was noted between the PCdST and PCdSNT treatments. However, the application of unmodified straw reduced plant-available Cd concentration by 43% and 63%, respectively, in CdSNT and PCdSNT, while the reduction observed in the modified material treatment was about 54% and 65%, respectively.

### 2.4. Phenanthrene Concentration in Soil

In the single phenanthrene treatment, the initial concentration of phenanthrene applied in soil was significantly reduced (*p* < 0.001) by 94% in the phenanthrene treatment applied with treated rice straw (PST) as compared to non-treated straw (PSNT), which recorded only 76% in reduction (Figure 4A). In the mixture treatment, the phenanthrene concentration in phenanthrene and Cd treatment applied with treated rice straw (PCdST) and non-treated straw (PCdNST) was more significantly reduced by 96% and 82%, respectively, compared to the control. This result suggests that chemical treatment could accelerate the removal of phenanthrene in soil.

Therefore, to better assess the impact of modified rice straw on degradation of phenanthrene in soil, soil samples contaminated with phenanthrene were only collected weekly to determine the contaminant concentration. The result showed that, within a week of incubation, 10% of phenanthrene applied in soil was reduced in PST, compared to 5% observed in PNST (Figure 4B). After 120 days, 97% of phenanthrene concentration was removed in PST compared to 88% and 77% recorded in PNST and the control treatment, respectively. This suggested that the modified rice straw might somehow be involved in phenanthrene removal in soil.

### 2.5. Predicted Functional Profile of Soil Metagenomes 

The presence of possible metabolites or others function involved in the metabolism and sorption of both organic and inorganic pollutants could highlight the reduction of pollutant availability and/or their total concentration in soil. Thus, a quantitative comparison of Clusters Orthologous Groups (COGs) among treatments soil samples was performed to identify functions among the treatment (Figure 5A). It was revealed that almost all the significantly changed individual COGs belonging to the categories of carbohydrate metabolism, ion transport, and energy metabolism were enriched in all treatments. In the single phenanthrene treatment, genes involved in carbohydrate metabolism were enriched remarkably compared to the control. However, the application of modified or unmodified straw did not manifest any significance difference regarding to functional diversity, with a similar Shannon index of 6.12 and 6.45, respectively. In the mixture treatment, the ion transport and ABC transporter involved in xenobiotic organic were significantly enriched in all treatments, but the trend was more pronounced (*p* < 0.01) in the treatment applied with rice straw either modified or unmodified. 

One other COG category enriched in phenanthrene treatment was molecular signaling. However, because the results gained for the COG categories disclosed a lack of congruency among the treatments, metagenomic sequences were allocated to the Kyoto Encyclopedia of Genes and Genomes (KEGG) database, which is particularly suitable for a comparison of metabolic pathways (Figure 5B). The comparison revealed that cytochrome P450, the metabolism of xenobiotic by cytochrome P450, and the drug metabolism cytochrome P450 were the most significant dominants in abundance. All of these genes were more enriched in the single phenanthrene treatment compared to the mixture treatment. This suggests that cytochrome P450 was involved in the degradation of phenanthrene.

### 2.6. Organic Products Involved in the Degradation Process

Figure 6 shows, in percentages, the different genes involved in the metabolism of organics materials identified in the different soil samples. Though the profiles of different treatments were similar, the abundance of genes involved in the phenanthrene single treatment was more pronounced in comparison with that in the mixture treatment. Genes involved in phtalic and salicylic acid metabolism, bisphenol, benzoate, and ethylbenzene degradation were more enriched in both treatments. This suggests that the enrichment of nearly all of genes involved in the degradation of different molecules in the single and mixture treatments with phenanthrene are linked to the degradation/transformation of phenanthrene. 

## 3. Discussion 

### 3.1. Rice Straw Structure

Agricultural crop residues, besides from the basic constituents, usually contained extractives, and several molecules composed by a variety of functional groups capable of binding heavy metal [24]. Studies have reported the use of NaOH to modify crop materials and improve metal sorption ability [30]. This could explain the profile variation and the frequency shifts between both rice straw samples (Figure 1). Additionally, a portion of silica was removed after the treatment of rice straw, leading to a significant variation of ST profile at 1039–400 cm^−1^. This suggests that after the removal of silica, some functional groups such as C-O, C-O-H, C-H, and O-H might became more exposed in the material. This is similar with the result found by Rocha et al., which reported that after alkali treatment the superficial layer of protective silica and the natural resins present in the rice straw were largely removed [21]. Li et al. showed that NaOH increased the O content, increased the surface basicity [31], and condensed organic matter to facilitate the subsequent activation of the materials [32]. Similarly, the evolution of pyrolysis product including C-H, C-O-H, and C-O-C functional groups were enhanced though chemical treatment; particularly, C=O was increased through NaOH treatment [33].

### 3.2. Rice Straw Decreased Cd Content in Sunflower Root and Shoot

The mobility and bioavailability of heavy metals are controlled by several factors and processes, including soil pH, soil organic matter content, metal form, adsorption, and the desorption process. Among them, soil pH plays a considerable role in the speciation and mobility of heavy metals, since its variation negatively correlates with metal availability and changes its impact on soil microorganisms and plants [23]. Indeed, at a low pH, organic and mineral soil materials tend to complex with metal to form stable metal-organo-mineral complexes. Conversely, when the pH is at a high level, this complex dissociates and ultimately increases the availability of the metal in the surrounding environment [23]. Thus, because of the addition of rice straw increased soil pH (Table 2), we can suppose that organic and mineral materials in the soil will associate with Cd^2+^ and form complexes in soil. Organic matter content is another key factor that can affect the bioavailability of Cd. Organic matter, in its dissolved form, can serve as a metals chelate and increase its mobility and uptake to plants; in its compact form, it constitute a major factor contributing to the soil ability to retaining metals in a non-exchangeable form [34]. The decomposition of rice straw and the chemical treatment led to the increase and exposure of certain functional groups, which can form complexes with Cd^2+^ (Figure 1). The metals cations have a good affinity with the anionic matters, such as COO^-^ and the phenolic group present on the rice straw surfaces, and can form stable complexes [25]. Park et al. showed that the sorption of Pb^2+^ in peanut straw was generally facilitated through complexation with the surface functional groups, especially carboxylic groups (COOH) [35]. This could explain why overall Cd content in plants was significantly reduced after rice straw addition (Figure 2). Therefore, if Pb^2+^ can be sorbed by COOH, other divalent metal cations such as Cd^2+^ could also react with the same groups in order to form stable complexes according to the following Figure 7. 

### 3.3. Effects of Rice Straw on CaCl_2_ and DTPA-Extractable Cd

Though CaCl_2_ and DTPA-extractable Cd methods used were well-adapted with a recovery rate of 96% ± 3, the CaCl_2_-extractable method showed a better correlation between the Cd plant accumulation and its extraction. This could be related to the increase in pH and organic matter content, which were reported to dramatically decrease DTPA-extractable heavy metals with their increase in soil [36]. The concentration of Cd extracted with both CaCl_2_ and DTPA was significantly low after the addition of rice straw (Figure 3). This can be explained by (i) the involvement of phosphorus concentration, (ii) the immobilization of Cd^2+^ through the increase in specific absorption, and (iii) the increase of electrostatic interaction and ionic exchange between Cd^2+^ and the anionic functional group. The addition of rice straw either as modified or unmodified material increased significantly the available phosphorus. This could explain the low concentration of Cd in both CaCl_2_ and DTPA extraction. A similar result was found by Cao et al., who showed that phosphorus concentrations in rice straw play an important role in the immobilization of heavy metals, such as Cd, Pb, and Zn [37], and reduce the metal translocation from the roots to the shoots by the formation or co-precipitation of insoluble metal phosphates in roots [38].

### 3.4. Effect of Rice Straw on Phenanthrene Concentration

PAHs are largely retained on the soil surface through organic partitioning of soil organic matter (SOM) which reduce their bioavailability [39]. The addition of rice straw increased the SOM content (Table 2) and reduced the phenanthrene concentration in soil (Figure 4). This could be attributed to the sorption capacity of SOM content and OC-normalized partitioning coefficient. It is nevertheless important to emphasize that phenanthrene, under precise conditions, can be sorbed by organic matter. However, the sorption coefficient K_oc_ may vary with the different types of organic matter, leading to uncertainties in predicting the environmental behavior of phenanthrene [40]. Apart from SOM, soil minerals can significantly contribute to the sorption of PAHs in the soil matrix [41] due to strong non-covalent interactions of their aromatic π- donor with sorbed cations on mineral surfaces [42].

### 3.5. Cd and Phenanthrene Interaction

In the dual treatment, both Cd and phenanthrene concentrations were significantly reduced in the soil. This suggests an interaction between the two chemicals. The most plausible explanation in this precise case would be the complexation of Cd^2+^ and phenanthrene, involving cation π-binding. Several studies have shown that the interaction between PAHs and divalent metal cations is related to cation- π-interaction [43]. Previously, it was found that, according to the chemical structure of both compounds, phenanthrene and Cd^2+^ could form a complex due to electrostatic interactions between quadripete benzenes of phenanthrene and positive charges of Cd [44].

### 3.6. Soil Microbial and Modified Rice Straw Affected Phenanthrene Concentration 

Though PAHs may undergo adsorption, microbial degradation is the major known mechanism to reduce phenanthrene concentration in soil. However, the extent and rate of degradation depends on several factors including pH, chemical structure, and the microbial population. The addition of modified rice straw increased not only soil microbial community, the abundance of 16S rDNA, and the bacterial Shannon index [45], it led to the exposition of some functional groups in the material (Figure 1). The prediction of the functional profile of soil biomes using COGs identified significant changes of individual COGs belonging to carbohydrate metabolism (including bisphenol, benzoate, and ethylbenzene degradation), and signaling pathway (including the cytochrome P450s categories (Figure 5 and Figure 6). This enlightened that the reduction of phenanthrene in the sampling soil was due to degradation or transformation. Figure 8 summarizes the possible mechanism related to how modified rice straw could promote phenanthrene degradation. Indeed, phenanthrene microbial degradation either in aerobic or anaerobic condition required interactions between enzymes and external substrate [46]. Microbial degradation may catabolize and breakdown phenanthrene through biotransformation into less complex metabolites or/and through aerobic/anaerobic mineralization into inorganic minerals; these processes may involve external subtracts such as bicarbonate molecules to activate their metabolism or the hydrogenation of the aromatic ring [18]. After chemical treatment, some functional groups such as C-O, C-O-H, C-H, and O-H became more exposed in the material and might be involved in the process of phenanthrene metabolism as external substrates. Thereafter, distinct microorganisms oxidized the carbon molecules through multi-enzyme complexes, such as ring-hydroxylating dioxygenase, that initiated the metabolism of phenanthrene by incorporating an oxygen molecule, resulting in a hydroxylated carbon to alcohol (dihydroxyphenathrene) [47] and the monooxygenase system of cytochrome P450 to catalyze the metabolic activation step of phenanthrene by the diol epoxide pathway, which is hydrolyzed to vicinal dihydrodiol phenanthrene (such as phenanthrene-9,10-dihydrodiol) [16,48]. Subsequently, the alcohol group is oxidized to aldehyde and, finally, to carboxylic acid [49]. Depending on the soil microorganism type, the oxidation of phenanthrene dihydrodiol to simple carbohydrate molecules may be undertaken either by adopting the phtalic acid route as an intermediate or by following the naphthalene route (salicylic acid as an intermediate) before being engaged in the central carbon cycle pathway [50,51].

## 4. Materials and Methods

### 4.1. Soil Properties

Soil was collected from the test field at Huazhong Agricultural University (HZAU) (30°28′26″N, 114°20′51″E), Wuhan, China, transferred to the greenhouse, air-dried, and sieved (2 mm) before being used. The soil used presented the following properties: pH (soil: H_2_O 1:2.5) 7.6; organic matter 1.31%; NH_4_Cl exchangeable K, 127.99 mg/kg; total nitrogen N 0.17%; Olsen-P of 39.69 mg/kg; CEC 11.47 cmol^+^/kg; and Ca, 2288.2 mg/kg 

### 4.2. Rice Straw Preparation

Rice straw of *Oryza sativa* (Taichung Native-1 variety) was collected at the agricultural field of HZAU, transferred to the laboratory, where it was washed three times with distilled water, dried at 60 °C, and crushed with grinder to obtain a fraction with particles size between 0.5–1 mm. The modified rice straw was prepared by adopting the procedure described in [21]. A sufficient quantity of triturated rice straw was dispersed in a HNO_3_ solution of 1.0 mol L^−1^ (1:5, m:v) and left agitating for 1 h at the room temperature. The content was after filtered and washed with water to remove the excess of acid in the material. Thereafter, the material was dispersed in a NaOH solution of 0.75 mol L^−1^ (1:10; m:v) and left under agitation for 1 h at the room temperature. The material once again was filtered, washed exhaustingly with water, and dried at 60 °C.

### 4.3. Chemicals Contamination 

Three kg of air-dried soil was placed into ceramic pots for the test experiment. A Cd concentration of 3 mg kg^−^^1^ was applied as cadmium chloride (CdCl_2,_ 98%, purity) solutions. This solution was poured on the soil surface, and the soil matrix was thoroughly mixed and incubated at 20 ± 1 °C for three months [52]. 

Phenanthrene (97% purity) dissolved in pure acetone (analytically pure) [53,54] was thoroughly mixed with the soil to produce a final concentration of 20 mg kg^−1^ of phenanthrene. The similar quantity of acetone only was used to prepare the control treatment with clean soil. The fresh contaminated soils were stored in open containers in a fume hood until all of the solvent evaporated [44]. In the chemical mixture treatment, the solution of 20 mg kg^−1^ of phenanthrene was added in the soil after three months of the incubation of the soil artificially contaminated by Cd and left for a week before being used for the experiment. 

### 4.4. Experimental Design

The experiment had nine treatments with three replicates, regrouped in three groups—including each single chemical contaminant and the mixture treatment. Each group had a control treatment (prepared with soil contaminated either by Cd^2+^ or phenanthrene only or by both Cd^2+^ and phenanthrene, respectively) and a treatment with unmodified (SNT) and modified (ST) rice straw for each group (Figure 9). Modified and unmodified straw (5.56 g/kg soil) was applied on the soil surface of the NST and ST treatments; the soil matrix was thoroughly mixed and then moistened with deionized water. This ratio was selected according to Memon previous study, which reported that this is the realistic ratio as a maximum rate of the residue incorporation under field conditions [55].

Five sunflower seedlings (*Helianthus annuus* L) were planted in pots and placed in the greenhouse. During the whole experimental period of 120 days, plants were watered daily and monitored weekly. On the 120 th day, the roots of sunflower were separated from their shoot and cleaned with water from piped supply to remove adhering soils before being washed with deionized water and dried at 60 °C for 7 days. 

### 4.5. Analytical Test

#### 4.5.1. Total Cadmium Determination

To conduct a total Cd in plant tissue analysis, double acid digestion of the sample was applied as described in [56]. Briefly, approximately 0.2 g of grinded plant sample was digested using 10 mL of a HNO_3_/HClO_4_ (4:1) mixture at 200 °C. The digested solution was diluted to 50 mL using deionized water and filtered before measuring Cd concentrations in an Atomic Absorption Spectrophotometry (AAS) (Z-2000, HITACHI, Tokyo, Japan). A standard material of plant GBW10015 (GSB-6) approved by general Administration of Quality Supervision, Inspection and Quarantine of the People’s Republic of China (AQSIQC) was used for the quality assurance and quality control (QA/QC) of Cd analytical procedure with recovery rate of 93 ± 7%.

#### 4.5.2. Extraction of Cd by CaCl_2_ and DTPA

The Cd concentration of the pore water was measured after extraction with 0.01M CaCl_2_ as described in [57]. Heavy metal extracted by DTPA in soil is considered as plant-available [58]. Thus, plant-available Cd was extracted from the treated soil using DTPA/TEA (pH 7.3) as described in [59]. The filtrates were subjected to AAS to determine Cd. The QA/QC for Cd in soil samples were estimated by determining Cd content in standard materials GBW07405 (GSS-5) purchased from National Central of Standard Materials in China, approved by AQSIQC with a recovery rate of 96% ± 3.

#### 4.5.3. Phenanthrene Determination 

The phenanthrene concentration in soil was determined by using the protocol described in our previous study [44]. The accuracy and analysis quality of phenanthrene measurements was estimated by determining phenanthrene content in the certified standard materials NIST1647 Priority Pollutant Polycyclic Aromatic Hydrocarbons in Acetonitril, purchased from Sigma-Aldrich with the recovery rate of 95 ± 3%.

#### 4.5.4. Fourier Transform Infrared Spectra of Rice Straw

FT-IR analysis of rice straw was performed as described in our previous study [23]. In brief, the freeze-dried rice straw sample (10 mg) was mixed with KBr (100 mg), then ground, homogenized, and pressed to reduce light scatter. Spectra were obtained by scanning the sample from 4000 to 400 cm^−1^ at 1 cm^−1^ resolution using Nicolet FTIR iS10 (ThermoFisher Scientific, Co., Ltd, Beijing branch, China)

### 4.6. DNA Extraction and Predicted Functional Profile

After the extraction of total DNA using a DNA isolation kit and the evaluation of quantity and quality by spectrophotometer, the soil bacteria abundance was estimated from qPCR assays targeting 16S genes in soil DNA extract. Illumina MiSeq sequencing and universal primers 515F (5′-GTGCCAGCMGCCGCGG-3′) and 806R (5′-GGACTACHVGGGTWTCTAAT-3′) were used to amplify the 16S rRNA V4 region for the analysis of soil microbial community at the Sangon Biotec Institute (Shanghai, China). The detail of the protocol used for the whole process can be found in [60]. 

To predict functional responses related to the degradation of phenanthrene in soil, PICRUSt [61] was used to generate a functional profile from 16S rRNA data. Prior to metagenomes prediction, the OTUs of 16S rRNA sequences were normalized by PICRUSt. PICRUSt, clusters of orthologous groups (COGs) [62], and the Kyoto Encyclopedia of Genes and Genomes (KEGG) were used to produce a table of functional genes, which were predicted to be present in the sample and to organize the genes into gene pathways respectively [63].

### 4.7. Statistical Analysis

All data were subjected to the ANOVA using SPSS statistical software, One and two-way ANOVA followed by Least significant Difference (LSD) with 95% confidence level were performed to assess the differences among means and multiple stepwise. A bioinformatics technique, PICRUSt, was used to explore the functional composition of that bacterial community. Network analysis was performed using the CoNet plugin for Cytoscape (http://www.cytoscape.org). Different graphs were drawn using Origin software

## 5. Conclusions

In the present study, it was found that modified rice straw could constitute an excellent bio-sorbent to remove or immobilize Cd^2+^ in soil, due to the increase of certain functional group including anionic matters groups which can form complexes with Cd^2+^ exposed on the modified rice straw surfaces. Rice straw also was found to influence soil characteristic which constitute keys factors for the mobility and bioavailability of heavy metals. In addition, the chemical structures of both pollutants may enhance them to interact and form a complex which can diminish the relative significance of the sorption of each pollutant. The modified rice straw, via interaction with soil microorganisms, might reduce phenanthrene concentration in soil. Our study offers clear and strong confirmation that modified rice straw could promote the reduction of heavy metal and the degradation of PAHs in soil. These findings suggest that, by the use of the forsaken natural resources, it is possible to improve the remediation of organic and inorganic pollutants in soil; this will permit a better understanding of soil biological and chemical interaction effects on metal and PAHs remediation technology.

## Figures and Tables

**Figure 1 ijms-20-02189-f001:**
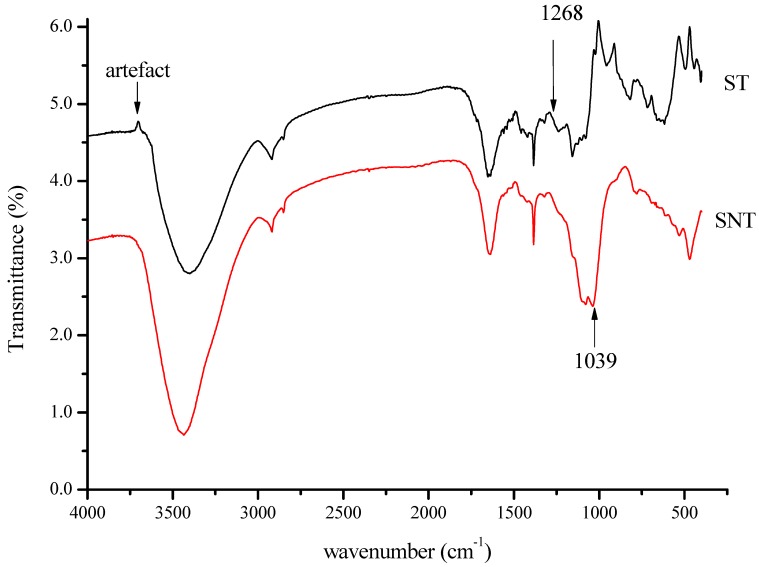
FT-IR in 4000–400 cm^−1^ spectra region of rice straw. SNT (straw not treated) indicates the spectra of rice straw untreated (natural), and ST (treated straw) indicates the spectra of rice straw treated with NaOH.

**Figure 2 ijms-20-02189-f002:**
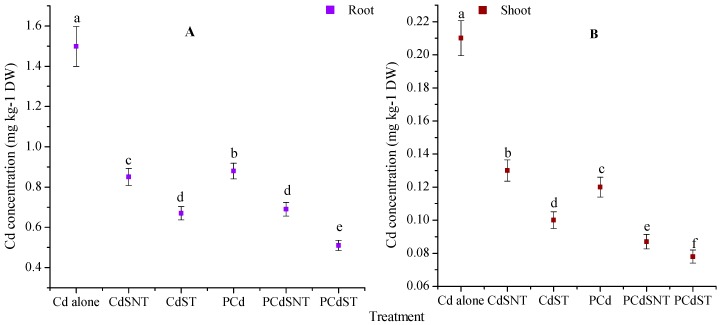
Effect of modified and unmodified rice straw on Cd concentration in the root (**A**) and shoot (**B**). In each group (**A** and **B**), the lowercase letters indicate a significant difference within different treatments (Tukey’tests, *p* < 0.001).

**Figure 3 ijms-20-02189-f003:**
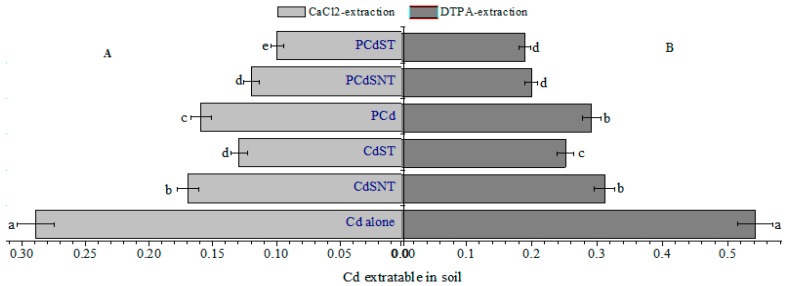
Effect of modified and unmodified rice straw on Cd extraction by CaCl_2_ (**A**) and diethylenetriaminepentaacetic acid (DTPA) (**B**) in soil. In each group (**A** and **B**), the lowercase letters indicate a significant difference within different treatments (Tukey’tests, *p* < 0.001).

**Figure 4 ijms-20-02189-f004:**
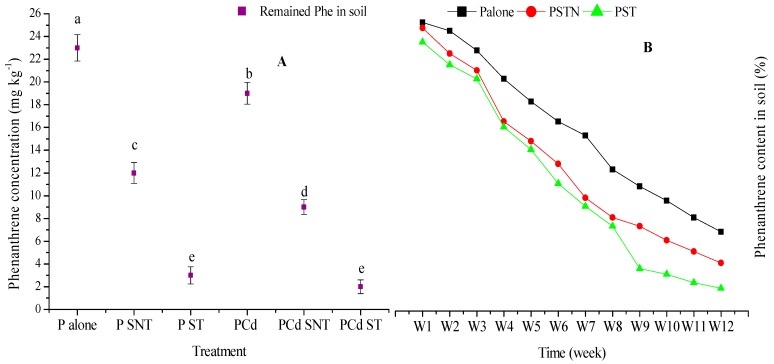
Effect of modified and unmodified rice straw on phenanthrene concentration in soil after 120 exposure days (**A**) and in each week during the 120 experimental days (**B**). Results present the mean of three replicates compared by Tukey’s tests. The lowercase letters indicate a significant difference within different treatments at *p* < 0.01 and *p* < 0.05 in A and B, respectively.

**Figure 5 ijms-20-02189-f005:**
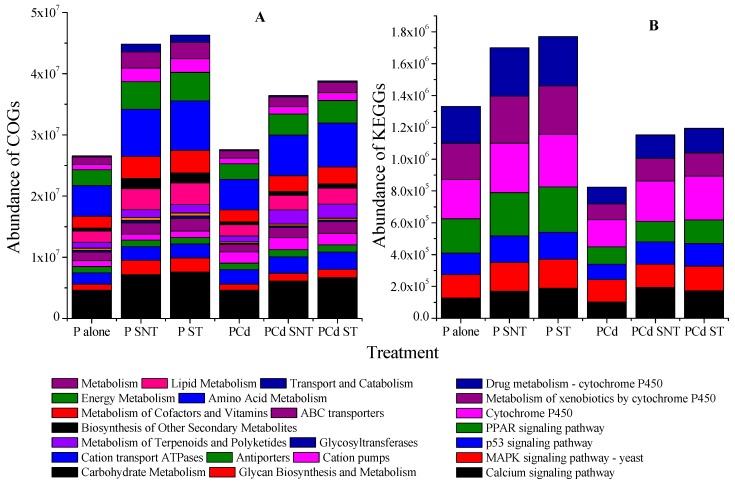
Effect of modified and unmodified rice straw on the distribution of Clusters Orthologous Groups (COGs) functional categories (**A**) and KEGGS metagenome functional prediction of the dataset (identified OTUs) (**B**).

**Figure 6 ijms-20-02189-f006:**
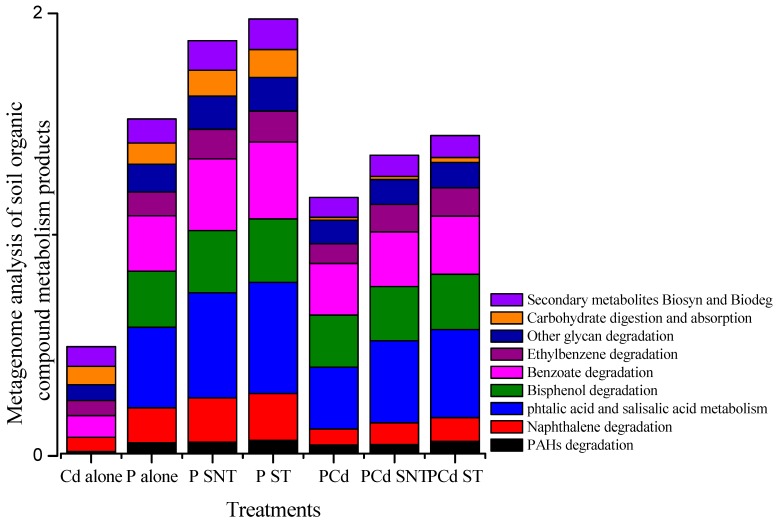
Effect of modified and unmodified rice straw on different genes involved in the metabolism of organics materials identified in the different soil samples (in percentage).

**Figure 7 ijms-20-02189-f007:**
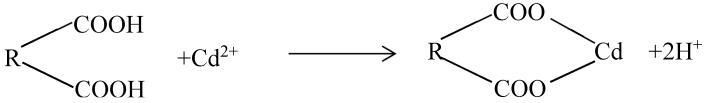
Sorbtion mechanism by which some functional group such as COOH complex with Cd2+ to immobilize it in soil.

**Figure 8 ijms-20-02189-f008:**
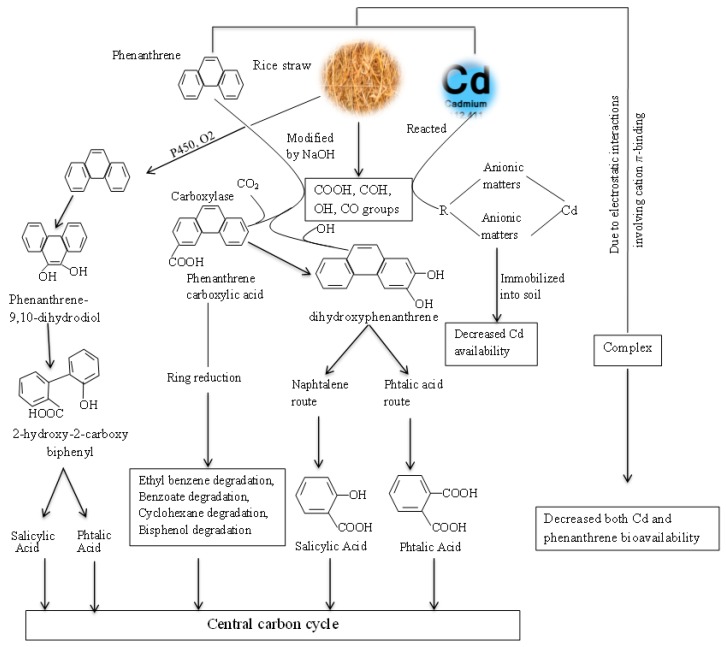
Schema summarizing the different possible mechanisms of rice straw to reduce Cadmium and Phenanthrene accumulation in sunflower plants and promote the degradation of phenanthrene in co-contaminated soil.

**Figure 9 ijms-20-02189-f009:**
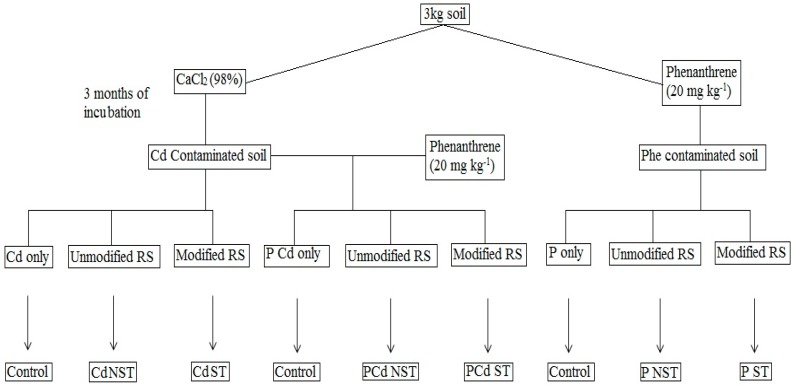
Summary of experimental design with all different treatments.

**Table 1 ijms-20-02189-t001:** Typical absorption bands and the main functional groups of rice straw.

Wave Number (cm^−1^)	Functional Groups	Compounds
3430	O-H stretching	cellulose and lignin
2919 and 2853	C-H stretching vibration	Aliphatic compounds
1640–1500	C=O	Ketone, carbonyl group
1450–1407	C=C stretching vibration	Aromatic skeletal
1388	C-H blending vibration	alkanes
1321–1302	C-O stretching and O-H blending	phenols, alcohols and esters
1268–1039	Si-O bending	
1242–1162	C-O-C stretching	aryl-alkyl ether
1070	C-O-C stretching vibration or C-O stretching and C-O deformation	ethanol group
1009	C-O-H and O-H blending	Decomposition of hemicellulose and cellulose
900–700	C-H	Aromatic hydrogen
700–400	C-C stretching	

**Table 2 ijms-20-02189-t002:** Influences of modified and unmodified rice straw on soil physical and chemical properties.

Treatment	CEC	pH	SOM	Total P	Available P
Cd alone	12.13	6.8	17.59	15.70	1.90
CdSNT	15.72	6.5	64.05 **	19.05	6.22 *
CdST	17.42 *	8.4 *	61.25 **	18.17	7.61 *
P alone	12.07	6.9	19.52	14.20	2.08
PSNT	15.54	6.3	68.52 **	18.42	6.41 *
PST	17.37 *	8.2 *	68.25 **	18.51	6.52 *
PCd alone	12.19	7.0	18.54	14.31	2.13
PCdSNT	16.15	6.4	71.52 **	16.38	7.2 *
PCdST	18.63 *	8.1 *	70.43 **	17.09	6.9 *

Data are the mean of three replicates tested by Two-way ANOVA analysis following by Tukey’s tests. The asterisks * and ** in the same column (nine values) indicate significant difference within treatments at *p* < 0.05 and *p* < 0.01, respectively. CEC indicates cation exchange capacity; SOM, soil organic matter and P, phosphorus.

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
