# Peer review of "Modified Rice Straw Enhanced Cadmium (II) Immobilization in Soil and Promoted the Degradation of Phenanthrene in Co-Contaminated Soil"

_ijms, 2019, doi:10.3390/ijms20092189_

Round 1
Reviewer 1 Report
Manuscript ID: ijms-468498
Title: Modified Rice Straw Enhanced Cadmium Immobilization in Soil and Promoted the Degradation of Phenanthrene in Co-Contaminated Soil
Reviewer comments
Technical note: line numbering isn’t continuous. The FTIR analysis is only skeletal and authors incorrectly identified band at 3704 cm-1 as -OH stretching band, while in fact it is an artifact with a negative absorption value. Authors list a “complete” set of “identified” bands in the Table 1 and left the interpretation to the reader.
At this stage the review of the manuscript is severely hindered by accumulation of acronyms which are not explained. While the meaning of some “generally accepted” acronyms can be deduced, authors did little to nothing to help guide the reader and the most basic acronyms used to describe the experimental procedures are also not explained. This is unacceptable practice.
The introduction should be re-written, because right now is nothing but confusing and lacking focus. It doesn’t explain the rational of the study and used methodologies.
Specific comments
Line 17: “on PAHs and heavy metal availability”. PAHs acronym not explained.
Line 19: “certain anionic matters groups on the modified rice straw surfaces, which can complex with Cd was exposed.” It is unclear what are you trying to convey in this sentence.
Line 21: “and DTPA extractable Cd, respectively.” DTPA acronym not explained.
Line 27: “exposed functional group might serve as external subtract and P450s as catalyzer”. Substrate? Catalyst?
Page 2, line 6: “bay-region and K-region epoxides which are highly reactive both chemically and biologically [3]. This is rather odd statement considering there is no oxygen atoms in phenanthrene molecule. Why only phenanthrene and cadmium were selected as a focus of the first paragraph of the introduction?
Page 2, line 13: “but also in many cases transfer the pollutant from one phase to another.” What these two phases would be?
Page 2, line 17: “Microbial degradation is the major technique mostly used to…” This is a new paragraph describing entirely different problem and should be formatted as such.
Page 3, line 7: “Although both spectra SNT and ST present”. Acronyms not explained.
Page 3, figure 1: The band marked as 3708 cm-1 in the spectrum of rice straw treated with NaOH (RST) is an artifact of data processing, where typically a CO2 background spectrum is subtracted and results in this kind of negative peaks. CO2 has several bands in the 3700-3500 cm-1 region. and an algorithm to remove these residual bands is typically implemented in the FTIR spectrometer data acquisition software.
Page 4, line 20: “The mean concentrations of Cd expressed as mg kg-1 DW” What is DW?
Page 4, line 22: “although the inoculation of rice straw significantly decreased Cd concentration”. What do you mean by “inoculation of rice straw”?
Page 4, line 26: “Further, in PCd treatment,” What is PCd?
Page 4, line 29: “respectively in PCdNST and PCdST treatment.” Acronyms not explained.
Page 7, line 4: “Thus, a quantitative comparison COGs among treatments soil samples”. Acronym not explained.
Page 12, line 25: “treatment with unmodified (SNT) and with modified (ST) rice straw for each group”. Acronyms not consistent with these used in figures 1, 5 and 6 (RSNT and RST).
Page 14, line 2: “FT-IR analysis of rice straw was performed as described in our previous study [21].” It is hardly acceptable reporting practice. A brief description is necessary, recapitulating used methodology.
Author Response
Thank you for your valuable comments. We have revised this manuscript according to your suggestions and comments. We hope that the revised manuscript would meet your requirements for publication in IJMS

Reviewer 2 Report
The manuscript entitled “Modified rice straw enhanced cadmium immobilization in soil and promoted the degradation of phenanthrene in co-contaminated soil” presents the investigation of the effect of umodified and modified rice straw on phenantrene and Cd availability in contaminated soil. This study offer confirmation that modified straw could promote the reduction of heavy metal and the degradation of phenantrene in soil.
The obtained results are interesting, but I recommend this manuscript to publication with major revision.
My comments are following:
1. Characterization of unmodified and modified rice straw is presented only by FTIR analysis. The authors should present more information concerning chemical composition of rice straw before and after modification, X-ray diffraction patterns or SEM images in order to better analysis of the effects of the modification.
2. The different possible mechanisms of rice straw to reduce Cd and phenathrene presented in manuscript were based only on literature data. Authors have not identified these products themselves in their studies. The identification of these products would greatly increase the value of this article.3. In the last sentence of the introduction it should be ..”(1) Cd and phenthrene availability;…” instead of “…(i) Cd and phenthrene availability;…”.
Table 1. Pearson’s correlation coefficient relating to phenthrene…” has wrong number, it should be : Table2. Pearson’s correlation coefficient relating to phenthrene…”.
Such an abbreviation as DTPA should be explained. Authors should describe how they obtain the values p.
Author Response

(The authors gave the same response as above.)

Round 2
Reviewer 1 Report
Manuscript ID: ijms-468498.R1
Title: Modified Rice Straw Enhanced Cadmium Immobilization in Soil and Promoted the Degradation of Phenanthrene in Co-Contaminated Soil
Reviewer 2 comments
This manuscript is a painful eyesore in its current form. I’m not sure why authors assumed after the first round of review it will be ‘easy’. The manuscript is barely readable: acronyms explained at the end, or not at all, 2) figure legends are minimalistic and don’t provide suitable description of statistical methods used, nor indicate which groups are compared and reached the statistical significance threshold, 3) Cadmium is referred to as Cd, despite it is Cd2+, and there is no reason to assume otherwise.
The style through the manuscript is atrocious and begs for grammatical corrections. Even technical terms are not used properly, and authors resort to technical slang or odd words with the meaning known only to the authors, e.g. “organo-mineral corps”.
The senior faculty should give proper guidance for the students/post-docs who were preparing the manuscript and spend some time to bring it to a reasonable level.
The section, describing the purity and the source of chemicals used in the study, is missing.
Specific comments
Line 19: “certain anionic matters groups” Just anionic groups.
Line 23: “metagenome using COGs” What are COGs?
Line 65: “it may become less toxic to the environment, or subsequently by shifting their valence by redox reaction by interacting in some specific cases with PAHs” Confusing description where a few topics of decontamination were included. Reduction of the Cd2+ does little to decrease the cadmium toxicity, so the focus should be on the removal techniques. There is no need to incorporate subjects which are trivial and tangent to the main subject.
Line 67: “However, to the best of our knowledge, this alternative is less exploited for the purpose of PAHs-heavy metal environmental depollution…” What is the other, standard method of Cd2+ removal? So, the PAHs play a dual role in this case, i.e., pollutant and complexation agent for the Cd2+, correct?
Line 85: “Rice straw was treated with NaOH solution to produce a carbonaceous…” Carbonaceous of what?
Line 100: “All cross-peaks were positive, implying that signals of functional groups did simultaneously intensity changes at the same direction with the treatment by NaOH.” What are you trying to say in this sentence?
Line 102: “of aliphatic materials.” Aliphatic materials?
Line 103: “there were displacements between both rice straw samples.” Displacements? These are frequency shifts or intensity changes. Use proper terminology. This is not a creative writing paper.
Line 107: “1039-400”. Missing units.
Line 107: “The changed profile of the absorption at 1039-400 for ST suggested that after chemical treatment and the removal of silica, the groups C-O, C-O-H, C-H and O-H became more exposed in the material.” You can’t draw these conclusions from these two poorly resolved and non-quantitative spectra.
Line 111: “Figure 1: FT-IR in 4000–400 cm−1 spectra region of rice straw. NRST indicates the spectra of rice straw untreated (natural); and RST the spectra of rice straw treated with NaOH”. NRST and RST acronyms are not present in the figure. Why this type of basic errors are present in the manuscript? It looks like somebody who was writing the legend didn’t look at the figure, which is just directly above. What is the meaning of the horizontal line terminated with two arrows on the top?
Line 113: “Table 1: Typical absorption bands…” Are authors planning to retain the first line of the strikethrough text (3708)? It is another example of sloppiness and lack of attention to detail.
Line 114: “2.2. Chemicals concentration” It is a terrible name for this section. What are you describing?
Line 116: “The mean concentrations of Cd expressed as mg kg-1 dry weight for the plant root and shoot in different treatments were investigated and plotted in Figure 2.” Is it a Cd or Cd2+? The same comment also applies to other occurrences of Cd in the manuscript. The phrase “expressed as mg kg-1 dry weight” is redundant, provided it is explained in figure 2 (or the legend).
Line 137: “The lowercase letters indicates significant difference at p < 0.001”. It is a meaningless description. The significance of p <0.001 is achieved for which groups? Instead of DW write “dry weight”.
Line 179: “The lowercase letters indicates a significant difference at p < 0.01 and p < 0.05 in A and B respectively.” It is another meaningless and non-informative description (similarly to fig. 1).
Line 233: “The mobility and bioavailability of heavy metals are controlled by the adsorption and the desorption process.” A generic statement, practically meaningless because it is missing the context of this work. Especially that it is followed by this statement: “Soil pH is one of the factors playing a considerable role in the speciation and mobility of heavy metals.” So, now the pH is important and not adsorption-desorption?
Line 237: “we can suppose that organo-mineral corps in the soil”. What are organo-mineral corps?
Line 252: “Table 2: Influences of modified and unmodified rice straw on soil physical and chemical properties.” None of the acronyms used in the table are explained. Specifically, what are: CEC, SOM, Total P, and Available P? And let’s not forget about the acronyms in the first column.
Line 321: “4.1. Soil proprieties”. What are proprieties? The title should be soil preparation, or similar.
Line 336: “Cd concentrations of 3 mg kg-1 were applied as cadmium chloride solutions.” It is not Cd; it is Cd2+. There is no need to repeat the Cd and cadmium in a single sentence. Rewrite for conciseness and accuracy of what was done.
Line 339: “was thoroughly spiked with the soil”. Slang (spiked).
Line 349: “treatment with unmodified (SNT) and with modified (ST) rice straw for each group”. Acronyms SNT and ST explained at the end of the manuscript. It looks like this manuscript was previously submitted (and rejected) to a journal with a typical section layout, i.e. the experimental section following the introduction, and authors moved the experimental at the end to match the layout of IJMS. Nothing wrong with that, except the logical order of acronym’s appearance, was not checked.
Line 374: “The QA/QC”. Acronym not explained.
Line 374: “Cd in soil samples were estimated by determine Cd content in standard materials”. Correct your English (estimated by determine).
Line 408: “In the present study, it was found that modified rice straw could constitute an excellent bio-sorbent to remove or immobilize Cd, due to the increase of certain anionic matters groups present on the rice straw surfaces which can complex with Cd and influence soil characteristic which constitute keys factors for the mobility and bioavailability of heavy metals.” It is one of the longest and unnecessarily convoluted sentences in the history of scientific writing. Please, try to express these thoughts in 2-3 well-thought sentences.
Line 411: “In addition, due to their chemical structure, the coexistence of Cd and phenanthrene diminished the relative significance of the sorption of each pollutant.” What is that mean?
Line 412: “The modified rice straw by interaction with soil microorganisms might reduce phenanthrene concentration in soil.” So, the microorganisms degrade the phenantrene and reduce its concentration. What is the role of rice straw? Might reduce? What are your results telling about that?
Author Response
Thank you for your valuable comments. We have revised this manuscript according to your suggestions and comments. We hope this time the revised manuscript would meet your requirements for publication in IJMS
Thank you for your valuable comments. We have revised this manuscript according to your suggestions and comments. We hope this time the revised manuscript would meet your requirements for publication in IJMS
General comment
This manuscript is a painful eyesore in its current form. I’m not sure why authors assumed after the first round of review it will be ‘easy’. The manuscript is barely readable: acronyms explained at the end, or not at all, 2) figure legends are minimalistic and don’t provide suitable description of statistical methods used, nor indicate which groups are compared and reached the statistical significance threshold, 3) Cadmium is referred to as Cd, despite it is Cd2+, and there is no reason to assume otherwise.
Response: once again thank you for the effort made to render this work presentable and scientifically correct. 1- All acronyms were explained as suggested. However, we wish to emphasize that, since most of the acronyms designate the different treatments used in this work, it is quite logical for them to be explained in the material and methods section. For information, there is even a figure (Figure 8) that summarizes the different treatments performed and explains all the acronyms that are not internationally known and that are specific to this work. But by the way, all acronyms were corrected and explained as suggested the reviewer. 2- the description of statistical methods used, and which groups are compared and reached the statistical significance threshold were made as underlined the reviewer. 3- In this manuscript it is true that cadmium is used as Cd, despite of Cd2+. Referring to the source by which cadmium is applied (CaCl2), it is evident that its inoculation was in ionic form based on the following reaction:
CaCl2 Cd2+ + 2Cl-
However, studies demonstrate that the hydrated radius Cd2+ (4.26 Å) has greater affinity for most of functional group in organic matter, including carboxylic and phenolic group which are hard Lewis bases. Although Cd2+ is a soft Lewis acid, its higher electronegativity (1.69) and pKH (negative log of hydrolysis constant) (10.1) ensure that Cd is favorable to adsorption through inner sphere surface complexation or sorption reaction in soil. Thus, in soil Cd is rarely found in his active and free form (Cd2+). Secondly, the digestion method used in the present manuscript to determine the concentration of cadmium either in soil or plant was “acid digestion”. Since in the soil, cadmium is trapped in the mineral and organic matrix, an oxidation during digestion is very important, because it allows to mineralize the organic matter by converting it in the form of water (H2O) and carbon dioxide (CO2). Thus, in order for the digestion of the organic matrix to be effective, the used acids should be strong inorganic acids whose purity is high as those used in this present study. This is why in most scientific publications dealing with similar topics uses Cd instead of Cd2+, unless the work is done in aqueous solution where the inert elements rate that can complex with Cd2+ is weak. Here are some examples: Zhao et al (2016), in a biological technique combining fungi and rice straw in an aqueous solution, used Cd2+ instead of Cd; Ding et al (2016) using alkali-modified biochar to remove Pb, Cu, Cd, Zn and Ni in aqueous solution also determined Cd2+ instead of Cd and Rocha et al (2009) who pointed out that rice straw would be a good biosorbent to remove Cd, Zn, Cu and Hg in an aqueous solution has also determined Cd2+ instead of Cd. Moreover, it is not surprising to find studies dealing with the subject on aqueous solutions and instead of determining Cd2+, authors uses Cd to express the concentration content or the absorption power of cadmium. This is the case of Park et al (2016) who studied the competitive absorption of heavy metals on sesame straw biochar in an aqueous solution. Inversely, research based on studies of metals in the soil use the atomic designation instead of their ionic form. This is the case, for example, of Huang et al (2018), which reported the binding characteristic of cadmium and copper to DOM derived from compost and rice straw, or Lu et al (2017), which demonstrated the biochar effect of bamboo and rice straw on the mobility and redistribution of Cd, Cu, Pb and Zn metals; or Meng et al (2017), who studied the contrasting effects of composting and pyrolysis on the bioavailability and speciation of Cu and Zn in pig manure.
The style through the manuscript is atrocious and begs for grammatical corrections. Even technical terms are not used properly, and authors resort to technical slang or odd words with the meaning known only to the authors, e.g. “organo-mineral corps”.
Response: the whole manuscript was checked and corrected
The senior faculty should give proper guidance for the students/post-docs who were preparing the manuscript and spend some time to bring it to a reasonable level.
Specific comments
1. Line 19: “certain anionic matters groups” Just anionic groups.
Response: not only anionic groups, there are of course other functional groups, but if we specify the ionic ones, it is just to emphasize according to what we would like to show in the present manuscript, but by the way, it was corrected in revised manuscript
2. Line 23: “metagenome using COGs” What are COGs?
Response: it was explained. Please see the revised manuscript
3. Line 65: “it may become less toxic to the environment, or subsequently by shifting their valence by redox reaction by interacting in some specific cases with PAHs” Confusing description where a few topics of decontamination were included. Reduction of the Cd2+ does little to decrease the cadmium toxicity, so the focus should be on the removal techniques. There is no need to incorporate subjects which are trivial and tangent to the main subject.
Response: it was corrected as suggested. Please see the revised manuscript
4. Line 67: “However, to the best of our knowledge, this alternative is less exploited for the purpose of PAHs-heavy metal environmental depollution…” What is the other, standard method of Cd2+ removal? So, the PAHs play a dual role in this case, i.e., pollutant and complexation agent for the Cd2+, correct?
Response: chemical precipitation, thermal processes, physical separation, phytoremediation involved compost, fly ash, biochar are other standard method to clean up and remove Cd2+ in the contaminated environment. According to studies, our previous works on the interaction of phenanthrene and cadmium on earthworms, and even the present study, phenanthrene could interact with cadmium and form complex which can reduce the availability of the two pollutant, so, to reply the question about phenanthrene play dual role: as pollutant and complexation agent, the response is yes. Even in the presence of phenanthrene the concentration of Cd was too much low compared to the treatment without it.
5. Line 85: “Rice straw was treated with NaOH solution to produce a carbonaceous…” Carbonaceous of what?
Response: Rice straw was treated with NaOH solution to produce carbonaceous molecules for the PAHs degradation activation and provide more binding sites for Cd. It was also corrected in the revised manuscript
6. Line 100: “All cross-peaks were positive, implying that signals of functional groups did simultaneously intensity changes at the same direction with the treatment by NaOH.” What are you trying to say in this sentence?
Response: Thank you for this question. What we would like to show here is that if all cross-peaks were positive, this would imply that signals of functional groups activated by NaOH did simultaneously intensity changes with each other at the same direction
7. Line 102: “of aliphatic materials.” Aliphatic materials?
Response: it was corrected. Please see the revised manuscript
8. Line 103: “there were displacements between both rice straw samples.” Displacements? These are frequency shifts or intensity changes. Use proper terminology. This is not a creative writing paper.
Response: it was corrected as suggested. Please see the revised manuscript
9. Line 107: “1039-400”. Missing units.
Response: it was corrected and added. Please see the revised manuscript
10. Line 107: “The changed profile of the absorption at 1039-400 for ST suggested that after chemical treatment and the removal of silica, the groups C-O, C-O-H, C-H and O-H became more exposed in the material.” You can’t draw these conclusions from these two poorly resolved and non-quantitative spectra.
Response: we understand that, as you say, from these two poorly resolved and non-quantitative spectra, conclusions are not appropriate. However scientifically it is quite reasonable to hypothesize or even to emit partial conclusions which, within the time they can be validated either by additional analyzes or by relying on similar works that found similar results. So, instead of concluding, we have corrected the active form into subjunctive one based on published scientific reports and results from this study, as a possible conclusion of what the result found suggested.
11. Line 111: “Figure 1: FT-IR in 4000–400 cm−1 spectra region of rice straw. NRST indicates the spectra of rice straw untreated (natural); and RST the spectra of rice straw treated with NaOH”. NRST and RST acronyms are not present in the figure. Why this type of basic errors are present in the manuscript? It looks like somebody who was writing the legend didn’t look at the figure, which is just directly above. What is the meaning of the horizontal line terminated with two arrows on the top?
Response: please accept our apologies, when we corrected the first revision, we did not realize mistakenly that we did not correct it. But in the present revised manuscript it was done. Please see the revised manuscript
12. Line 113: “Table 1: Typical absorption bands…” Are authors planning to retain the first line of the strikethrough text (3708)? It is another example of sloppiness and lack of attention to detail.
Response: please once again accept our apologies, but actually if you find that it was kept in the table, it was not our fault, because during the revision, it was deleted, but maybe the journal editorial office did not make attention on the modification in that place, the proof is that even in Figure 1 it was mentioned artefact instead of the wavenumber which was initially found there
13. Line 114: “2.2. Chemicals concentration” It is a terrible name for this section. What are you describing?
Response: it was corrected and changed. Please see the revised manuscript
14. Line 116: “The mean concentrations of Cd expressed as mg kg-1 dry weight for the plant root and shoot in different treatments were investigated and plotted in Figure 2.” Is it a Cd or Cd2+? The same comment also applies to other occurrences of Cd in the manuscript.
Response: As we said in the begging, even the whole mechanism to explain different phenomena and results from this present study concern and use the active form and free cadmium Cd2+, the determined concentration here was not for Cd2+, but rather that of Cd. Since the study was performed in soil area, cadmium was trapped in the mineral and organic matrix, thus after acid digestion mineral and organic matter was converted in the form of water (H2O) and carbon dioxide (CO2) and cadmium was determined as Cd concentration, as most of similar studies did
15. The phrase “expressed as mg kg-1 dry weight” is redundant, provided it is explained in figure 2 (or the legend).
Response: it was corrected as suggested. Please see the revised manuscript
16. Line 233: “The mobility and bioavailability of heavy metals are controlled by the adsorption and the desorption process.” A generic statement, practically meaningless because it is missing the context of this work. Especially that it is followed by this statement: “Soil pH is one of the factors playing a considerable role in the speciation and mobility of heavy metals.” So, now the pH is important and not adsorption-desorption?
Response: it is obvious that the initial paragraph was not well organized, several basic information were missed, it is why the reader or reviewer could be confusing about the importance of each factors and process. In the present revised manuscript, we have do our best to correct and give a clear explication about what we would like to say in the initial manuscript. Please see the revised manuscript
17. Line 237: “we can suppose that organo-mineral corps in the soil”. What are organo-mineral corps?
Response: to the best of our knowledge, in scientific writing, sometime some specific words can be combined to express a meaning which converge together and avoid a repetition. For example: “chemical” and “physical” characteristic can be combined in “physicochemical” characteristic. It is the same situation used in the present case; “organic” and “mineral” material can be combined as “organo-mineral material”. What was wrong here maybe is the term “corps” which express the same mining as ‘element’. However to avoid any other misunderstanding of the manuscript, it was corrected as a simple way as suggested. Please see the revised manuscript
18. Line 252: “Table 2: Influences of modified and unmodified rice straw on soil physical and chemical properties.” None of the acronyms used in the table are explained. Specifically, what are: CEC, SOM, Total P, and Available P? And let’s not forget about the acronyms in the first column.
Response: thank you to underline this case. However we find that all these pointed acronyms are all explained below the table. CEC indicates cation exchange capacity; SOM, soil organic matter and P, phosphorus. Others acronyms still the same ones used in the whole manuscript, no other new one was added, except the fact that maybe they were written in a wrong way. But as I was asked to be rechecked, it was corrected in a good way. Please see the revised manuscript
19. Line 321: “4.1. Soil proprieties”. What are proprieties? The title should be soil preparation, or similar.
Response: actually the tittle “soil properties” was used because that section was made to show the used soil properties. But because we would like to avoid repetition and plagiarism with our previous study, we preferred to orientate readers to our published papers containing soil characteristics instead of mentioning them. However, to make the writing of the manuscript in a scientific logic and to avoid misunderstanding, we have reported the missing information to this section in the present revised manuscript. Please see the revised manuscript.
20. Line 336: “Cd concentrations of 3 mg kg-1 were applied as cadmium chloride solutions.” It is not Cd; it is Cd2+. There is no need to repeat the Cd and cadmium in a single sentence. Rewrite for conciseness and accuracy of what was done.
Response: actually the repetition of Cd and cadmium in the single sentence can be explain in this way: the first one written as Cd is the main subject of the sentence, but the second one written as “cadmium” is not only cadmium but cadmium chloride which constitute one thing (CdCl2): the source used to contaminated soil by Cd
21. Line 339: “was thoroughly spiked with the soil”. Slang (spiked).
Response: it was corrected as suggested. Please see the corrected manuscript
22. Line 349: “treatment with unmodified (SNT) and with modified (ST) rice straw for each group”. Acronyms SNT and ST explained at the end of the manuscript. It looks like this manuscript was previously submitted (and rejected) to a journal with a typical section layout, i.e. the experimental section following the introduction, and authors moved the experimental at the end to match the layout of IJMS. Nothing wrong with that, except the logical order of acronym’s appearance, was not checked.
Response: Of course, it can be a possibility; but not necessarily that it was the case. Most of the scientific redaction and the standard requirement of scientific writing use the IMRD (Introduction, Material and Methods and Discussion). Yes, the first draft was written by adopting the IMRD plan, then when we decided to submit the manuscript in IJMS, it was necessary to arrange the manuscript according to the journal requirement. We do not see how could we explained all the acronyms in another section, when normally the experimental design is in the Material and Methods section. Since most of the acronyms designate the different treatments used in this work, it is quite logical for them to be explained in the material and methods section. There is even a figure (Figure 8) that summarizes the different treatments performed and explains all the acronyms that are not internationally known and that are specific to this work.
23. Line 374: “The QA/QC”. Acronym not explained.
Response: these acronyms were explained just before the bracket where they were mentioned for the first time. Please see the revised manuscript
24. Line 374: “Cd in soil samples were estimated by determine Cd content in standard materials”. Correct your English (estimated by determine).
Response: it was corrected as suggested. Please see the revised manuscript
25. Line 408: “In the present study, it was found that modified rice straw could constitute an excellent bio-sorbent to remove or immobilize Cd, due to the increase of certain anionic matters groups present on the rice straw surfaces which can complex with Cd and influence soil characteristic which constitute keys factors for the mobility and bioavailability of heavy metals.” It is one of the longest and unnecessarily convoluted sentences in the history of scientific writing. Please, try to express these thoughts in 2-3 well-thought sentences.
Response: it was corrected and expressed in better way as suggested. Please see the revised manuscript
26. Line 411: “In addition, due to their chemical structure, the coexistence of Cd and phenanthrene diminished the relative significance of the sorption of each pollutant.” What is that mean?
Response: what we would like to say here is that the chemical structure of both pollutants may enhance them to interact and form a complex which diminished the relative significance of the sorption of each pollutant.
27. Line 412: “The modified rice straw by interaction with soil microorganisms might reduce phenanthrene concentration in soil.” So, the microorganisms degrade the phenantrene and reduce its concentration. What is the role of rice straw? Might reduce? What are your results telling about that?
Response: the last section in the Discussion summarize the interaction effect of modified rice straw and soil microorganism. It is not evident to separate both rice straw and soil microbial for many reason. Until proven otherwise, Microbial degradation is the major technique mostly used to remove, alter, or isolate PAH in a contaminated environment. However, the degradation necessitates others factors to support the process, such as pH, physical and chemical structure and the microbial population, specific microorganism which can produce specific enzyme which are involved in the metabolism process, and moreover external substrates which are used for the activation of the metabolism. Thus the modified rice straw increased not only soil microbial community, the abundance of 16S rDNA and bacterial Shannon index, but also led to the exposition of some functional groups in the material that might be involved in the process of phenanthrene metabolism as external substrate. The functional profile of soil biomes identified significantly change on carbohydrate metabolism including bisphenol, benzoate and ethylbenzene degradation and signaling including cytochrome P450s categories which are also involved in in phenanthrene metabolism
References:
1- Zhao, M-H., Zhang, C-S., Zeng, G-M., Cheng., M and Liu., Y. 2016. A combined Biological removal of Cd2+ from aqueous solution using Phanerochaete chrysosporium and rice straw. Ecotoxicology and Environmental Safety
2- Ding, Z-H., Hu, X., Wan, Y-S., Wang, S-S., Gao, B. 2016. Removal of lead, copper, cadmium, zinc, and nickel from aqueous solutions by alkali-modified biochar: Batch and column tests. Journal of Industrial and Engineering Chemistry
3- Rocha, C. G.; Zaia, D. A. M.; da Silva Alfaya, R. V.; da Silva Alfaya, A. A. 2009. Use of rice straw as biosorbent for removal of Cu (II), Zn (II), Cd (II) and Hg (II) ions in industrial effluents. Journal of hazardous materials
4- Park, J.-H.; Ok, Y. S.; Kim, S.-H.; Cho, J.-S.; Heo, J.-S.; Delaune, R. D.; Seo, D.-C. 2016. Competitive adsorption of heavy metals onto sesame straw biochar in aqueous solutions. Chemosphere
5- Huang, M., Li, Z., Huang, B., Luo, N., Zhang, Q., Zhai, X. and Zeng, G., 2018. Investigating binding characteristics of cadmium and copper to DOM derived from compost and rice straw using EEM-PARAFAC combined with two-dimensional FTIR correlation analyses. Journal of hazardous materials. 344, 539-548.
6- Lu, K.; Yang, X.; Gielen, G.; Bolan, N.; Ok, Y. S.; Niazi, N. K.; Xu, S.; Yuan, G.; Chen, X.; Zhang, X. 2017. Effect of bamboo and rice straw biochars on the mobility and redistribution of heavy metals (Cd, Cu, Pb and Zn) in contaminated soil. Journal of environmental management, 186, 285-292
7- Meng, J., Wang, L., Zhong, L-B., Liu, X-M., Brookes, P-C., Xu, J-M. 2017. Contrasting effects of composting and pyrolysis on bioavailability and speciation of Cu and Zn in pig manure. Chemosphere.
Reviewer 2 Report
Corrections are ok
Author Response
Thank you for to review our manuscript and provide us valuable comments
Round 3
Reviewer 1 Report
Manuscript ID: ijms-468498.R2
Title: Modified Rice Straw Enhanced Cadmium Immobilization in Soil and Promoted the Degradation of Phenanthrene in Co-Contaminated Soil
Reviewer 2 comments
There is no such thing like “internationally recognized acronyms”, and if these are not provided in the manuscript it hinders the evaluation. Even for the sake of the readers who are unfamiliar with this specific area, these should be explained.
Specific comments
Line 2: “Modified Rice Straw Enhanced Cadmium…” It should be Cadmium(II). You’re not using zerovalent cadmium in your experiments.
Line 19: “…using phenanthrene and cadmium (Cd) as model.” You’re not using zerovalent cadmium (Cd) as a model, but the solution of Cd2+ (or cadmium(II)), and these two are internationally recognized symbols for divalent cadmium. The same applies to all the instances where Cd is used incorrectly in the manuscript, whenever the Cd2+, a soluble form of cadmium is described.
Line 420: “followed by Least significant Different (LSD)”. It is called Least Significant Difference.
Author Response
Thank you for your valuable comments. We really appreciated your aptitude and the manner you have used to review our manuscript. your comments and suggestions were more than instructive not only for us, but we are very sure that they increased the value and scientific quality of our different manuscript that you review and as well as the ours. We hope this time the revised manuscript would meet your requirements for publication in IJMS.
God bless you!!!!!
